# Optimizing Efficiency and Motility of a Polyvalent Molecular Motor

**DOI:** 10.3390/mi13060914

**Published:** 2022-06-09

**Authors:** Mark Rempel, Eldon Emberly

**Affiliations:** Department of Physics, Simon Fraser University, 8888 University Dr, Burnaby, BC V5A 1S6, Canada; eemberly@sfu.ca

**Keywords:** molecular motor, burnt bridge ratchet, computational model

## Abstract

Molecular motors play a vital role in the transport of material within the cell. A family of motors of growing interest are burnt bridge ratchets (BBRs). BBRs rectify spatial fluctuations into directed motion by creating and destroying motor-substrate bonds. It has been shown that the motility of a BBR can be optimized as a function of the system parameters. However, the amount of energy input required to generate such motion and the resulting efficiency has been less well characterized. Here, using a deterministic model, we calculate the efficiency of a particular type of BBR, namely a polyvalent hub interacting with a surface of substrate. We find that there is an optimal burn rate and substrate concentration that leads to optimal efficiency. Additionally, the substrate turnover rate has important implications on motor efficiency. We also consider the effects of force-dependent unbinding on the efficiency and find that under certain conditions the motor works more efficiently when bond breaking is included. Our results provide guidance for how to optimize the efficiency of BBRs.

## 1. Introduction

Transporting bio-molecules is an essential process in living things. At short length scales, diffusion is effective but at long distances it gives way to active, energy burning transport processes involving molecular motors. Due to the highly ordered and complex nature of living systems, entropic processes must be countered by processes that consume useful energy in favor of mechanical work. Typically, chemical energy in the form of ATP is used to provide the free energy needed for these motors to overcome diffusion and other disordering mechanisms [1].

Molecular motors convert chemical energy into mechanical work using several mechanisms. Motors such as kinesin and myosin use an active power stroke, whereby chemical energy initiates a conformational change in the motor that can exert a force on its surroundings. Other motors such as collagenase, ParAB and RNAzyme use a burnt bridge ratchet (BBR) mechanism, in which they interact with a patterned substrate and catalyze its removal, burning a path and rectifying their motion [2,3,4,5]. These BBR motors are suggested to fulfil a variety of tasks such as the faithful partitioning of DNA, transport of cellular proteins and the motility of viruses along a cell surface [6,7,8,9,10,11,12,13]. BBRs utilize the formation of motor-substrate bonds (bridges) and the bridges’ subsequent destruction (burning) along with random spatial fluctuations to facilitate sustained directed motion.

A growing body of experimental and theoretical work has helped to uncover the physical and chemical mechanisms that drive the motion of BBRs. Several reconstitution efforts have demonstrated the viability of synthetic BBRs. The organisation and motion of synthetic cargo was demonstrated in vitro by utilizing a gradient of MinD and the diffusophoetic effect [14]. Vechiarelli et al. reconstituted the SopABC system and demonstrated that the motility could be adjusted by altering system parameters such as chemical kinetics and substrate concentration [15]. Using DNA–RNA binding, another cell-free study by Blanchard et al. demonstrated that motors could move against large persistent forces and interact with thousands of substrate molecules simultaneously [16,17]. Substantial work has also been conducted to produce in vitro BBRs in which the bond strength for each substrate-motor bond could be altered by adjusting the motor-substrate binding energy [15,16,18]. By adjusting the bond strength it is possible to tune the rate of force-dependent unbinding. Additionally, ParAB inspired models have shown that the substrate concentration is proportional to the motor’s steady-state speed and the speed could be optimized by altering substrate to motor interaction lengths [4].

The above work has focused on the motility of BBRs and their optimization—their speed, persistence and processivity. However, these systems are driven far from equilibrium and have a constant source of chemical energy that powers their motion, and so, how can one optimize their efficiency? In motors such as kinesin and F1-atpase that have an active power stroke, the efficiency is well characterized, reaching as high as 40–80% and nearly 100%, respectively [19,20,21]. With respect to BBRs, they rectify thermal motion without any power stroke, and so, by definition, will be less efficient. Nevertheless, it has not yet been established how a BBR’s efficieny depends on its chemistry and force generation. For instance, will a BBR be more efficient at low or high burn rates, and is it better to have a high or low concentration of substrate?

In this paper, we explore how to optimize the efficiency of a polyvalent motor hub undergoing burnt bridges ratcheting using a deterministic model. The motor binds substrate and catalyzes its removal at some burn rate. From the motion of the motor and its rate of substrate consumption we calculate its Stokes efficiency, which is the ratio of the output power (against the drag force) to the rate of chemical energy input [22]. We find that for a fixed amount of substrate, the optimal efficiency occurs at an intermediate substrate consumption rate. We also consider force-dependent unbinding of the motor from the substrate, and find that under certain conditions it is favourable to have some unbinding for more efficient operation. Thus, it is possible that a motor which burns every single substrate it comes in contact with can operate less efficiently than one which can unbind substrate. Our findings lead to testable predictions, in particular, for in vitro synthetic burnt bridge ratchets where both the binding and unbinding kinetics can be tuned.

## 2. Materials and Methods

In our model, a polyvalent motor hub interacts with substrate molecules that are attached to a surface of polymers (see Figure 1A), or are themselves polymers such as in the DNA-RNAse-based systems. We assume that the hub’s motion is confined to one dimension. The substrate can exist in one of three states: (i) surface-bound, (ii) bound to the hub or (iii) free in the buffer (Figure 1A). (This extends our previous model that did not explicitly consider the dynamics of the substrate-hub complex) [4,23]. The concentration of surface-bound substrate on the track at position *X* is $A_{s}\left(X\right)$. The hub can bind to surface-bound substrate and form a complex leading to a concentration of bound substrate at *X* given by $A_{c}\left(X\right)$ (Figure 1B). Substrate in the complex can be burnt and released back into the buffer where it is in the free state and has a concentration $A_{f}$ that is independent of position since we assume that the buffer is well mixed. Substrate in the buffer can bind to the surface at a rate, $k_{on}$, and can also unbind at a rate $k_{off}$ (Figure 1C). The system is closed and starts with an initial substrate concentration, $A_{tot}$. At all future times the substrate concentration satisfies, $\int A_{tot}dX=\int A_{f}\left(t\right)dX+\int A_{s}\left(X,t\right)dX+\int A_{c}\left(X,t\right)dX$.

When the hub is bound to the substrate, the complex experiences a linear restoring force due to the entropic elasticity of the underlying surface to which the substrate is attached. The total force from the bound complex on the hub, located at $X_{h}$, is then,
$$F=\int -k\left(X_{h}-X\right)A_{c}\left(X\right)dX\;,$$
where the entropic elasticity is given by the effective spring constant, $k=\frac{k_{B}T}{\sigma^{2}}$, and σ is the characteristic fluctuation length of the polymers and $k_{B}T$ is thermal energy. The dynamics of the hub are in the over-damped regime and so the elastic force is balanced by the drag force. The equation of motion for the hub is,
(1)$$\gamma \frac{dX_{h}}{dt}=-\int k\left(X_{h}-X\right)A_{c}\left(X\right)\;dX\;,$$
where γ is the viscous drag coefficient. For a spherical particle moving through bulk, this may be approximated as $\gamma =6\pi \eta^{\prime }R$, where $\eta^{\prime }$ is fluid viscosity and *R* is the radius of the particle. However, due to hydrodynamic effects near boundaries, this expression breaks down [24]. Nevertheless, we assume the hub is constantly tethered to the surface during its motion and is therefore at a constant height. So the drag coefficient is constant, albeit larger than what it would be in bulk solution.

As the surface polymers fluctuate, the surface-bound substrate moves either further or closer to the hub. Those that come in contact with the hub form a bound complex at rate, *r*. The energy of the fluctuation is $\frac{1}{2}k{\left(X-X_{h}\right)}^{2}$; assuming that these fluctuations are in equilibrium compared to the timescale of the motion of the hub, the rate of complex formation obeys Arrhenius kinetics with $r\exp\left[-\frac{\beta }{2}k{\left(X-X_{h}\right)}^{2}\right]A_{s}\left(X\right)$. The hub then catalyzes the dissociation of substrate in the bound complex to the free state at a burn rate, ν. For now, we will assume that there is no unbinding of bound complex back into the surface-bound state; this will be added later. The chemical kinetics for the bound complex using the above assumption is,
(2)$$\frac{dA_{c}\left(X\right)}{dt}=re^{-\frac{\beta }{2}k{\left(X-X_{h}\right)}^{2}}A_{s}\left(X\right)-\nu A_{c}\left(X\right)\;.$$

The dynamics of surface-bound substrate has several contributions, including the loss due to the formation of the bound complex. We assume that it can unbind and return to the buffer at a rate $k_{off}$ and free substrate in the buffer can bind to the surface at a rate $k_{on}$. We assume that we are in the limit where the overall effect of the hub on the concentration of surface-bound and free substrate is minimal, and that the buffer has come to steady state with the surface. Under these assumptions the binding rate, $k_{on}A_{f}$ at steady state is,
$$k_{on}A_{f}=\frac{k_{off}}{\left(1+k_{off}/k_{on}\right)}A_{tot}$$
(See Appendix A info for full derivation).

The full chemical kinetics of the surface-bound substrate then follows,
(3)$$\frac{dA_{s}\left(X\right)}{dt}=\frac{k_{off}}{\left(1+k_{off}/k_{on}\right)}A_{tot}-k_{off}A_{s}\left(X\right)-re^{-\frac{\beta }{2}k{\left(X-X_{h}\right)}^{2}}A_{s}\left(X\right)\;.$$

Equations (Equation 1)–(Equation 3) can be non-dimensionalised using the substitutions X=σx, $t=\frac{\tau }{r}$ and $A_{i}\left(x\right)=a_{0}a_{i}\left(x\right)$, leading to the following dimensionless equations,
(4)$$\frac{dx_{h}}{d\tau }=-\int \left(x_{h}-x\right)a_{c}\left(x\right)\;dx\;,$$
$$\frac{da_{c}\left(x\right)}{d\tau }=e^{-\frac{{\left(x-x_{h}\right)}^{2}}{2}}a_{s}\left(x\right)-\nu^{\prime }a_{c}\left(x\right)\;,$$
$$\frac{da_{s}\left(x\right)}{d\tau }=\frac{k_{off}^{\prime }}{\left(1+K\right)}a_{tot}-k_{off}^{\prime }a_{s}\left(x\right)-e^{-\frac{{\left(x-x_{h}\right)}^{2}}{2}}a_{s}\left(x\right)\;,$$
where $\nu^{\prime }=\frac{\nu }{r}$, $K=\frac{k_{off}}{k_{on}}$ and $k_{off}^{\prime }=\frac{k_{off}}{r}$. We define the reference concentration to be $a_{0}=\gamma r\sigma /k_{B}T$. (Note that this reference concentration depends on the size of the hub through, γ, its ability to bind substrate through, *r*, and the characteristic fluctuations of the underlying surface, σ. Changing any of these at a fixed substrate concentration is effectively equivalent to changing the substrate concentration). Thus, the dynamics of the hub depends on four dimensionless parameters: the total substrate concentration, $a_{tot}$, the hub induced substrate burn rate, $\nu^{\prime }$, the substrate equilibrium constant or substrate turnover rate, *K*, the free unbinding rate of substrate and $k_{off}$. Table 1 provides an overview of the characteristic quantities used for non-dimensionalisation. We assume that the chemical kinetics governing substrate binding to the surface is similar to its binding with the hub and so for simplicity, we set $k_{on}=r$. Thus, $K=k_{off}^{\prime }$ in our simulations.

### 2.1. Force-Dependent Unbinding

In the above, we made the assumption that all substrate bound to the hub could never unbind and would eventually get burnt. Here, we relax that assumption and allow the complex to disassociate back to its surface-bound state at a rate, ω. We also allow for this to have an additional force dependence due to the restoring force pulling on the substrate-hub bond. This is parameterized by Δ, the distance separating the bound complex and surface-bound states. The energy difference between the bound complex and surface-bound state due to the work performed by the restoring force is,
$$\Delta E=-F\left(X\right)\Delta =-k|X_{h}-X|\Delta \;.$$

The kinetics for the bound complex and surface-bound substrate become,
$$\frac{dA_{c}\left(X\right)}{dt}=re^{-\frac{\beta k}{2}{\left(X_{h}-X\right)}^{2}}A_{s}\left(X\right)-\nu A_{c}\left(X\right)-\omega e^{\beta k|X_{h}-X|\Delta }A_{c}\left(X\right)\;,$$
$$\frac{dA_{s}\left(X\right)}{dt}=\frac{k_{off}}{\left(1+k_{off}/k_{on}\right)}A_{tot}-k_{off}A_{s}\left(X\right)-re^{-\frac{\beta k}{2}{\left(X_{h}-X\right)}^{2}}A_{s}\left(X\right)+\omega e^{\beta k|X_{h}-X|\Delta }A_{c}\left(X\right)\;.$$

These can be non-dimensionalised by following the same substitutions as above in addition to setting $\Delta =\frac{\delta }{\sigma }$ and $\omega =\omega^{\prime }r$. to yield,
(5)$$\frac{da_{c}\left(x\right)}{d\tau }=e^{-\frac{1}{2}{\left(x_{h}-x\right)}^{2}}a_{s}\left(x\right)-\nu^{\prime }a_{c}\left(x\right)-\omega^{\prime }e^{|x_{h}-x|\delta }a_{c}\left(x\right)\;\mathrm{and}$$
(6)$$\frac{da_{s}\left(x\right)}{d\tau }=\frac{k_{off}^{\prime }}{\left(1+K\right)}a_{tot}-k_{off}^{\prime }a_{s}\left(x\right)-e^{-\frac{1}{2}{\left(x_{h}-x\right)}^{2}}a_{s}\left(x\right)+\omega^{\prime }e^{|x_{h}-x|\delta }a_{c}\left(x\right)\;.$$

We solve Equations (Equation 4)–(Equation 6) using a 4th order Runge–Kutta method with adaptive stepping. The ODE solver package solve_ivp from scipy.integrate was used to find both the trajectory of the motor and the time evolution of the substrate concentration. The spatial coordinate was descretized with spacing of ΔX=0.1σ.

### 2.2. Efficiency

The efficiency of a motor is defined to be the ratio of output work (or power) to input work (or power). Many motors operate between two baths at different temperatures, where the optimal efficiency is given by the Carnot efficiency. Biomolecular motors operate in a bath that is at a fixed temperature and so the flows of heat and work arise from and to the same bath. In the low Reynolds regime, the efficiency is the ratio of power conducted on viscous drag force over the rate of chemical energy usage [22]. This ratio is known as the Stokes efficiency and defined as,
(7)$$\eta =\frac{\gamma V^{2}}{B}\;.$$
where *V* is the velocity of the hub and *B* is the rate of change in chemical free energy [27]. For our model we calculate $B=\Delta G\nu \int A_{c}\left(X\right)dX$ where ΔG is the free energy change per substrate molecule burned. The systems we consider are far from chemical equilibrium where the amount of product is small compared to consumable substrate. Thus, ΔG is many $k_{B}T$ and drives the conversion of substrate into product, which when coupled mechanically to the hub, can be used to do mechanical work. We wish to non-dimensionalise Equation (Equation 7). Setting $\Delta G=\Delta gk_{B}T$ and making the same space, time and concentration substitutions as before, we get the non-dimensionalised numerator of Equation (Equation 7),
$$\gamma V^{2}=\gamma \sigma^{2}r^{2}v^{2}.$$ Here, v=V/σr is the non-dimensionalised velocity. Similarly for the denominator,
$$B=\Delta G\nu \int A_{c}\left(X\right)dX=\Delta gk_{B}Ta_{0}\sigma r\nu^{\prime }\int a_{c}\left(x\right)dx.$$ Substituting each of these into Equation (Equation 7) and recognizing that $\gamma \sigma^{2}r^{2}/k_{B}Ta_{0}\sigma r=1$ we arrive at a form of the Stokes efficiency in terms of non-dimensionalised quantities,
(8)$$\eta =\frac{1}{\Delta g}\frac{v^{2}}{\nu^{\prime }\int a_{c}\left(x\right)dx}=\frac{1}{\Delta g}\frac{v^{2}}{b}\;.$$ Here, $b=\nu^{\prime }\int a_{c}\left(x\right)dx$ is the non-dimensionalised substrate consumption rate.

Table 2 provides an overview of system variables and typical values.

## 3. Results

Here, we present the results of our deterministic model for a polyvalent hub on a substrate with total substrate concentration $a_{tot}$ and with the hub burning the substrate at a rate $\nu^{\prime }$ (see Figure 1 for schematic).

Figure 2A shows representative snapshots of a hub as it moves across a substrate and how the formation of bound complex establishes a surface-bound substrate gradient. In order to break symmetry and initiate motion, we set the initial concentration of the surface-bound substrate to be $\frac{a_{tot}}{1+K}$ in front with a triangular well behind. There is an initial transient period where bound complex builds up and the hub’s speed changes, but at later times the hub reaches a steady-state speed and a traveling wave-front is formed in the surface-bound substrate. Figure 2B shows the time dependence of the velocity of the hub at different burn rates, $\nu^{\prime }$, for a fixed total substrate concentration, $a_{tot}=1.0$. The initial spike in velocity results from a surplus of surface-bound substrate available to the hub. At some burn rates, after the initial transient spike in speed, the hub eventually reaches a constant steady-state speed. At very high and very low burn rates the hub cannot set up a sustainable chemical gradient and its speed gradually decays. In Figure 2C, we show the consumption rate, *b*, that also peaks and reaches steady state at similar time scales to the velocity. Notably, an increase in $\nu^{\prime }$ does not necessitate an increase in *b*; this is due to fact the that consumption rate scales with velocity. The more surface-bound substrate the hub moves through the more substrate it consumes.

In Figure 3, we show the resulting steady-state velocity and consumption rate as a function of $\nu^{\prime }$ and $a_{tot}$. For a fixed $a_{tot}$, an optimal value of $\nu^{\prime }$ exists that maximizes the velocity (Figure 3A). A $\nu^{\prime }$ that is small will allow for the hub to attach to the substrate but the lack of burning does not allow for a significant gradient to form, while a large $\nu^{\prime }$ quickly burns off any bonds with the substrate and again no sustained gradient is formed to drive motion. For both large and small $\nu^{\prime }$, the lack of a significant chemical gradient leads to low velocity but the complete stalling results from the rebinding of substrate to the surface that fills in the wake behind the hub, removing any gradient that had existed (Figure 3A inset i,iii). If the burn rate facilitates a substantial gradient, the hub may outpace the substrate rebinding and maintain its chemical gradient (Figure 3A inset ii). In this case, the majority of bound complex exists in front of the hub providing a constant forward force. For fixed, $\nu^{\prime }$, increasing $a_{tot}$ leads to an increase in velocity (Figure 3B) since additional available substrate increases the force applied to the hub. The velocity saturates due to an ever increasing wake (Figure 3B inset): leftover surface-bound substrate at the rear of the hub. A large wake allows for binding of substrate behind the hub, resulting in an increased backward force.

In Figure 3C, we show the consumption rate of the motor as a function of the burn rate. For a given $a_{tot}$, the consumption rate is maximal when the velocity is greatest (Figure 3C). This can be understood as the hub moving through more surface-bound substrate allowing it to consume more of it. Figure A1 shows the velocity as a function of consumption rate; we show that the curves are closely approximated by linear equations, emphasising that b∼v. Regardless of the velocity of the motor, the consumption rate is always greater than zero because of the continual rebinding of surface-bound substrate. When the velocity of the hub is zero the consumption rate is independent of burn rate; this is especially clear in Figure 3D where consumption rate is plotted as a function of substrate concentration, $a_{tot}$. On a log-log plot, when v=0, the slope of the curve is 1 showing that consumption depends linearly with $a_{tot}$. When the hub velocity is non-zero, the consumption rate changes, but eventually, when the hub speed saturates consumption returns to scale linearly with $a_{tot}$. We also show that the consumption rate scales with $a_{tot}$ by considering the steady-state surface-bound substrate concentration, $a_{s}^{*}$, by setting Equation (Equation 3) to zero. Ignoring the effects of the motor and force-dependent unbinding, at steady state, $a_{s}^{*}=\frac{1}{1+K}a_{tot}$ (see Appendix A for derivation) indicating that $a_{s}^{*}$ only depends on the total substrate concentration and the substrate turnover rate. From Equation (Equation 2), at steady state $e^{\frac{{\left(x-x_{h}\right)}^{2}}{2}a_{s}^{*}}=\nu^{\prime }a_{c}^{*}$, therefore, $b∼a_{tot}$.

Using our results for the hub’s velocity and the burn rate of the substrate, we calculate the Stokes efficiency of our hub as a function of the chemistry. For fixed $a_{tot}$, there is an optimal efficiency at a specific $\nu^{\prime }$ (see Figure 4A). Slow and fast burn rates lead to a stalled motor and hence zero efficiency. The peak efficiency occurs at the same $\nu^{\prime }$ that leads to the highest velocity. This can be understood, since at steady state the consumption rate b∼v, so $\eta ∼v^{2}/v=v$. Any $\nu^{\prime }$ that generates a peak in *v* will also generate a peak in efficiency for a fixed $a_{tot}$. Figure 4B shows how the efficiency varies with changing initial substrate concentration, $a_{tot}$, for a fixed burn rate, $\nu^{\prime }$. For a given $\nu^{\prime }$, there is an optimal substrate concentration, $a_{tot}$, that leads to the highest efficiency. Since the consumption rate scales with $a_{tot}$, the efficiency scales with $\frac{1}{a_{tot}}$, so a substrate concentration that is too large will lead to inefficient operation.

In Figure 4C, we plot the efficiencies as a function of $\nu^{\prime }$ and $a_{tot}$ as a heat map. Although one can continue to increase the steady-state speed of the hub by increasing $a_{tot}$ and $\nu^{\prime }$, the same is not true for efficiency. Indeed for a given substrate binding chemistry, there is is an optimal value of $a_{tot}$ and $\nu^{\prime }$ that leads to the greatest efficiency. When operating at high speeds, the hub is burning relatively more substrate, leading to lower efficiencies. At the opposite end of low $a_{tot}$ and $\nu^{\prime }$, the hub is effectively stalled, though it continues to burn substrate (due to rebinding of substrate), yielding zero efficiency. At optimal $\nu^{\prime }$ and $a_{tot}$, the hub moves fast enough to maintain a substrate gradient but slow enough that the consumption rate remains small.

We now show how the unbinding rate of the substrate, $k_{off}^{\prime }$, affects efficiency. The effects of substrate turnover, from the surface to the buffer and back, decrease with $k_{off}^{\prime }$. A small $k_{off}^{\prime }$ means that little surface-bound substrate is liberated into the buffer, and with low concentrations of free substrate in the buffer, substrate rebinding is slow. This effect is also seen mathematically as every rebinding term in Equation (Equation 6) scales with $k_{off}^{\prime }$. In Figure 5, we show that the overall efficiency of the motor decreases with greater $k_{off}^{\prime }$. This is due to a larger wake caused by a high substrate turnover. A larger wake creates a large amount of negative work on the motor, lowering the efficiency. Substrate is still consumed but motion is opposed. In addition to this, the optimal efficiency shifts to lower values of $\nu^{\prime }$ and $a_{tot}$ as $k_{off}^{\prime }$ decreases. Because there is lower turnover of substrate, each substrate molecule can produce useful work, and at low speeds, the hub makes use of every substrate interaction allowing it to be very efficient. For completely irreversible substrate binding (i.e., $k_{off}^{\prime }=0$), our findings show that efficiency increases with decreasing concentration and burn rate. However, at low enough substrate concentrations and burn rates, stochastic effects become important which make the assumptions leading to our deterministic formulation problematic. We expect that stochastic effects at low substrate concentrations and burn rates would favour diffusive behaviour of the motor, and hence inefficient operation. For large $k_{off}^{\prime }$ (e.g., $k_{off}^{\prime }=0.1$), there is high substrate turnover leading to a rapid filling in of the wake, and so the hub must operate at larger $a_{tot}$ and $\nu^{\prime }$ in order to move. However, this comes at the cost of being less efficient.

### Motor Efficiency Including Force-Dependent Complex Dissociation

We now consider the situation where substrate bound to the complex can dissociate before being burned. This is parameterized by a complex dissociation rate, $\omega^{\prime }$. We will also consider that this dissociation is force-dependent due to the pulling force acting on the substrate-hub bond. This is parameterized by δ, the distance between the bound and unbound states of the substrate to the hub (δ=0 corresponds to no force dependence). In Figure 6A, we show the normalized velocity of the hub with a fixed $\nu^{\prime }=1.0$ as a function of dissociation rate at several different substrate concentrations, $a_{tot}$. The velocities are normalized by the largest velocity for each curve. At high substrate concentrations the velocity stays near the maximal velocity as $\omega^{\prime }$ increases while it drops away more rapidly for smaller $a_{tot}$. However, the overall effect of including dissociation of substrate from the hub is to reduce the velocity of the motor. In Figure 6B, the normalized consumption rate is plotted as a function of $\omega^{\prime }$. Not surprisingly, the consumption rate goes down with increasing dissociation rate. It decreases most rapidly at lower values of substrate concentration. Figure 6C shows the resulting Stokes efficiency as a function of $\omega^{\prime }$. At low substrate concentrations the peak efficiency occurs when $\omega^{\prime }=0$. Interestingly, at higher substrate concentrations, the peak efficiency is at $\omega^{\prime }\neq 0$. Thus, having some amount of complex dissociation back into free substrate leads to more efficient motor operation. We can make sense of this by recalling that at higher $a_{tot}$ the consumption rate decreases while little change in velocity occurs. Figure 6D shows the effect of including force-dependent dissociation. For a fixed $a_{tot}$ and $\nu^{\prime }$, increasing δ simply shifts the efficiency curve. At larger δ, the force acting on the hub leads to a greater overall dissociation rate. The result is that efficiency is maximized at a lower overall unbinding rates, $\omega^{\prime }$, when force is included.

## 4. Discussion

In this paper, we explored the efficiency of a polyvalent motor hub acting as a burnt bridge ratchet using a minimal deterministic model for its chemistry and dynamics. Through computational simulations we mapped out the dependence of the hub’s velocity and consumption rate as a function of the total substrate concentration, $a_{tot}$ and burn rate, $\nu^{\prime }$. We found that the motor exhibited steady-state ballistic motion in regions of the parameter space and stalling in others. As found in previous studies, for a fixed amount of substrate, there is an optimal burn rate, $\nu^{\prime }$, that leads to the highest motor velocity. We found that at a fixed burn rate, $\nu^{\prime }$, increasing $a_{tot}$ led to an increase in velocity up to a point where it saturated. This was due to the increasing concentration of unburnt substrate left behind the hub that could balance any increases in force from the front. Using the speed and consumption rate, we found that the maximal efficiency for a given $a_{tot}$ is located at the maximal velocity while the maximal efficiency for a given $\nu^{\prime }$ is located well below its maximal speed. This is because the consumption rate scales with $a_{tot}$ and therefore the efficiency scales with the $\frac{1}{a_{tot}}$. As *v* saturates with increasing substrate concentration, the efficiency stops increasing and is dominated by the large $a_{tot}$ and begins to decrease.

We also looked at the effect of varying $k_{off}^{\prime }$ on the efficiency. We found that the maximal efficiency decreases with greater $k_{off}^{\prime }$ due to the negative work that results from large amounts of substrate turnover. A large substrate turnover allows the system to return to the steady-state substrate concentration faster. We also saw that for $k_{off}^{\prime }$ that discourages substrate turnover, the efficiency global maximum was located at low $a_{tot}$ with correspondingly low $\nu^{\prime }$. In the low turnover regime, optimal efficiency is higher at low velocities as a result of efficiency scaling as $\frac{1}{a_{tot}}$; despite the decreased hub velocity, the decreased $a_{tot}$ dominates the efficiency. Furthermore, the optimal $a_{tot}$ and $\nu^{\prime }$ were much higher when substrate turnover became significant. When substrate turnover is significant, the motor requires enough substrate to allow it to move faster and outpace the substrate rebinding.

Finally, we saw that in some regions of the $\nu^{\prime }$/$a_{tot}$ parameter space that some degree of force-dependent unbinding may provide local improvement to efficiency. It was found that the bound complex to surface-bound substrate transition rate, $\omega^{\prime }$, could improve efficiency by decreasing the burn rate. We also noted that the distance between the bound complex and surface-bound states, δ, shifts the optimal $\omega^{\prime }$; δ should be accounted for when optimizing efficiency. We expect that molecular motor engineers may make use of some force unbinding to improve their motors’ efficiency and should not always attempt to reduce the force unbinding to zero.

This work highlights key parameters that can be tuned to optimize the efficiency of synthetic BBR motors. In many cases, the substrate concentration, burn rate and force-dependent unbinding rate can be controlled by the experimenter. The substrate concentration has been controlled in several previous experiments [15,16]. In some cases, the burn rate may be adjusted by altering the concentration of enzymes selected to catalyze the motor-substrate bonds [16]. The force-dependent unbinding rate has been tuned in other experiments by affecting the free energy associated with the binding [15,16,18]. Future work can verify these results in vitro and utilize them to optimize motor efficiency. The testing of in vivo systems may also reveal how well tuned living systems are to optimize efficiency and motility.

## Figures and Tables

**Figure 1 micromachines-13-00914-f001:**
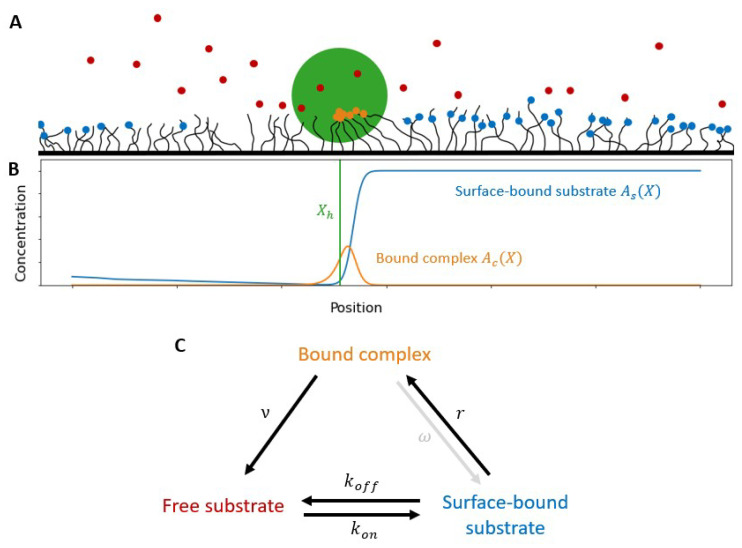
Model schematic. (**A**) The motor hub (green) interacts with surface-bound substrate (blue) forming bound complex (orange) which is eventually burnt and freed to the buffer (red). (**B**) Example concentration surface-bound substrate and bound complex and the relative location of the hub at a single point in time. (**C**) Schematic of state transitions and associated rates. The bound complex to surface-bound state transition initially ignored.

**Figure 2 micromachines-13-00914-f002:**
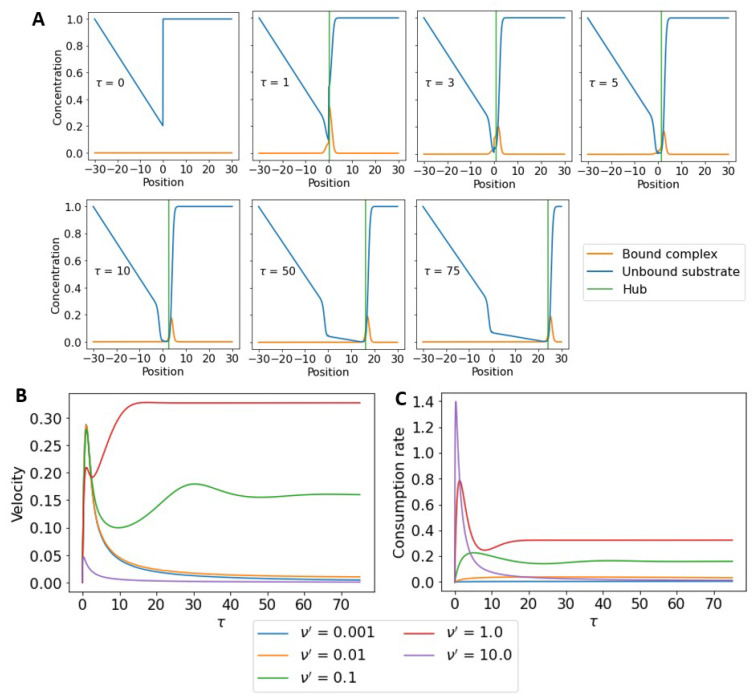
Time dependence of hub velocity and substrate consumption rate. (**A**) Substrate concentration profiles and the hub position at different points in time. (**B**) The velocity of the motor as a function of time at different burn rates, $\nu^{\prime }$. There is an initial transient period where the hub accelerates and attains peak speed. For some burn rates, the hub eventually achieves a constant steady-state velocity. Whereas at very high or very low burn rates, there is a steady slowing down of the hub with time. (**C**) The consumption rate, *b*, as a function of time at different burn rates. The substrate consumption rate peaks and reaches steady state at similar times to the velocity. In all of the above calculations, $a_{tot}=1$, $\omega^{\prime }=0$, $k_{off}^{\prime }=K=0.01$.

**Figure 3 micromachines-13-00914-f003:**
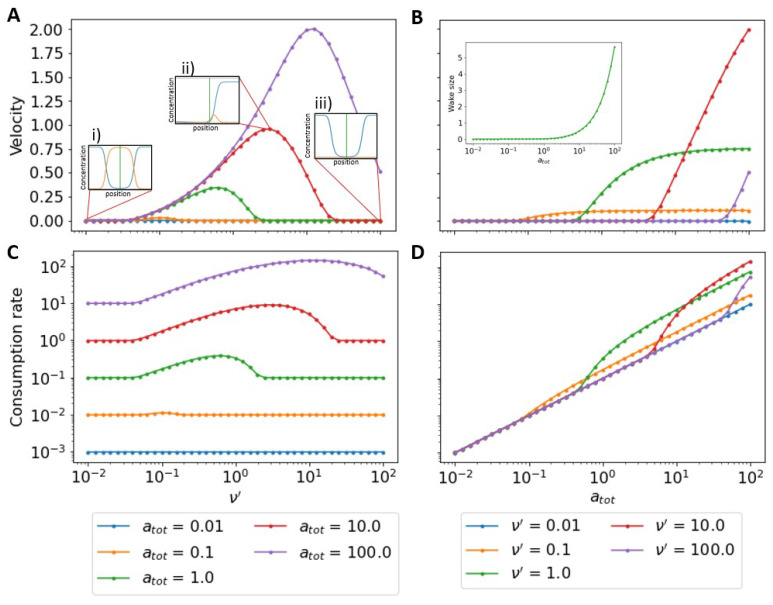
Dependence of steady-state velocity and consumption rate on system parameters. (**A**) Steady-state velocity as a function of $\nu^{\prime }$ at different values of $a_{tot}$. At fixed $a_{tot}$ there is an optimal $\nu^{\prime }$ that leads to the largest hub speed. (**B**) Steady-state velocity as a function of $a_{tot}$ at different $\nu^{\prime }$. At fixed $\nu^{\prime }$, the steady-state velocity increases with $a_{tot}$, but eventually saturates. (**C**) Steady-state substrate consumption rate as a function of $\nu^{\prime }$ at fixed $a_{tot}$. As with velocity, the consumption rate peaks at a given burn rate. (**D**) Steady-state consumption rate as a function of $a_{tot}$ for fixed $\nu^{\prime }$. At a fixed burn rate, the consumption rate increases without bounds with increasing $a_{tot}$. In all of the above calculations, $\omega^{\prime }=0$, $k_{off}^{\prime }=K=0.01$.

**Figure 4 micromachines-13-00914-f004:**
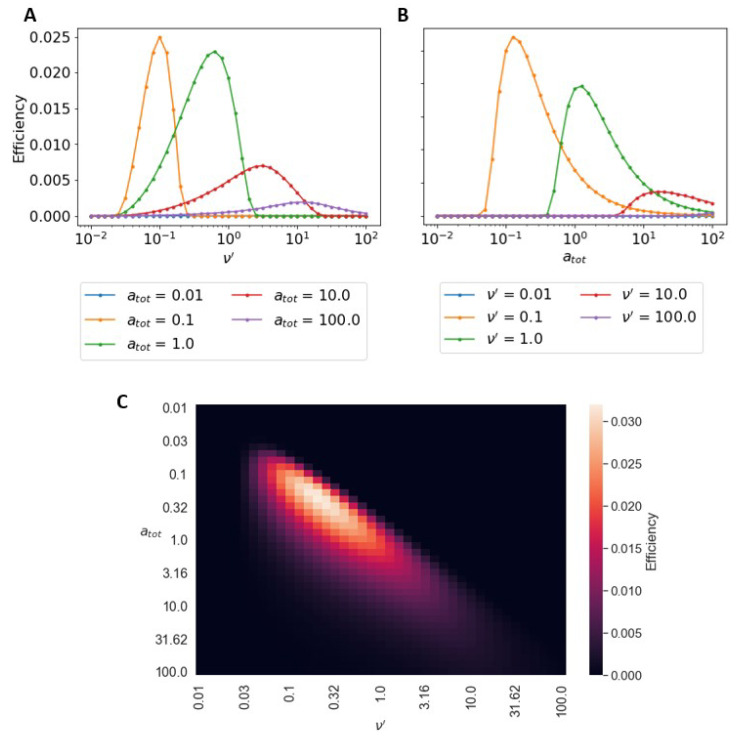
Dependence of efficiency on system parameters with no complex unbinding. (**A**) Stokes efficiency of system as a function of burn rate, $\nu^{\prime }$ at different substrate concentrations $a_{tot}$. Given $a_{tot}$, $\nu^{\prime }$ may be adjusted to provide maximum efficiency. Optimal efficiency coincides with $\nu^{\prime }$ that maximizes velocity. (**B**) Efficiency as a function of $a_{tot}$ at different burn rates, $\nu^{\prime }$. (**C**) Heat map of the efficiency as a function of $a_{tot}$ and $\nu^{\prime }$. There is an optimal $a_{tot}$ and $\nu^{\prime }$ that lead to the highest efficiency given a fixed amount of input chemical energy, Δg. In all of the above calculations, $\omega^{\prime }=0$, $k_{off}^{\prime }=K=0.01$, Δg=15.

**Figure 5 micromachines-13-00914-f005:**
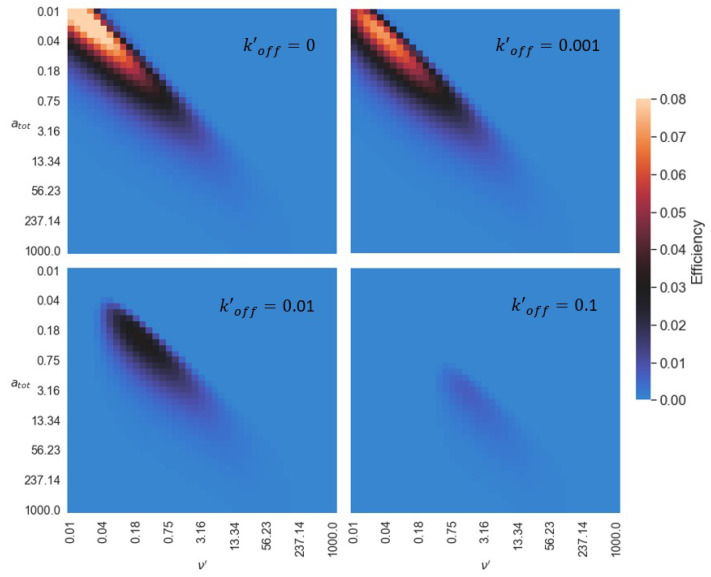
Effects of rebinding on efficiency. Efficiency heat maps for different $k_{off}^{\prime }$. In all of the above calculations, $\omega^{\prime }=0$, Δg=15.

**Figure 6 micromachines-13-00914-f006:**
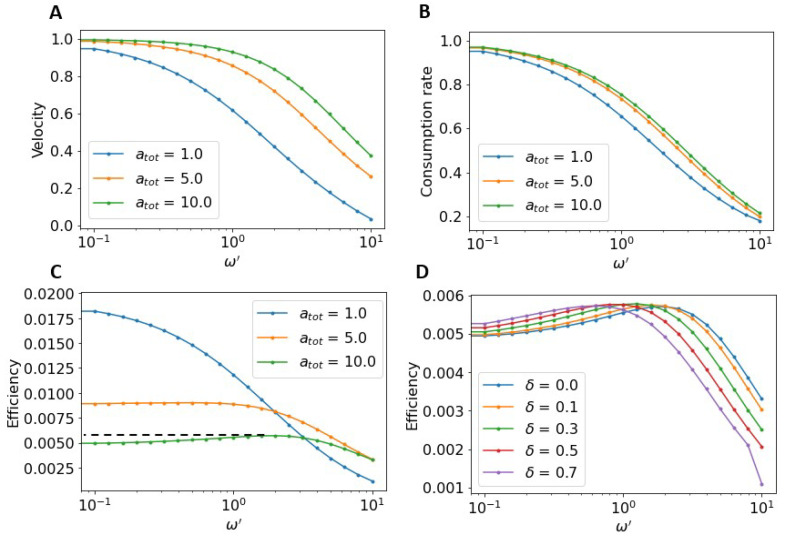
Effects of complex unbinding on efficiency. (**A**) The normalized velocity as a function of $\omega^{\prime }$. Velocities normalized by the maximum velocity for a given $a_{tot}$. (**B**) The normalized consumption rate as a function of $\omega^{\prime }$. (In A,B some curves approach 1 slowly). (**C**) The absolute efficiency as a function of $\omega^{\prime }$ at different $a_{tot}$. At lower substrate concentrations the maximal efficiency occurs when there is no complex unbinding (i.e., $\omega^{\prime }=0$). At higher substrate concentration (i.e., $a_{tot}=10$) the optimal efficiency occurs at a non-zero unbinding rate. For calculations in (**A**–**C**), δ=0. (**D**) The efficiency as a function of $\omega^{\prime }$ at different δ. For larger δ the optimal $\omega^{\prime }$ is greater. δ makes no significant difference in the global maximal efficiency of the motor. For calculations in D, $a_{tot}=10$. In all of the above calculations, $\nu^{\prime }=1$, $k_{off}^{\prime }=K=0.01$, Δg=15.

**Table 1 micromachines-13-00914-t001:** Characteristic Quantities.

Quantity	Characteristic Quantity	Symbol	Typical Value	Non-Dimensionalised Quantity
time	complex formation rate	*r*	20 s${}^{-1}$ [17]	τ=rt
length	characteristic fluctuation	$\sigma =\sqrt{k_{B}T/k}$	100 nm ^†^	x=X/σ
concentration	reference concentration	$a_{0}=\gamma r\sigma /k_{B}T$	10 μm${}^{-1}$ ^‡^	$a_{tot}=A_{tot}/a_{0}$

^†^ Calculated using using $k_{B}T=4.1$ pN nm and k∼0.2 pN/μm [25]. ^‡^ Calculated using $\gamma =6\pi \eta^{\prime }R$, where $\eta^{\prime }=10^{-3}$ Pa  s [26] and R∼1 μm [16]. For reasons mentioned earlier, this value for the drag coefficient should be viewed as a first order approximation.

**Table 2 micromachines-13-00914-t002:** Model Variables.

Variable	Definition	Typical Value	Non-Dimensionalised Quantity
β	inverse thermal energy	0.2 pN${}^{-1}$ nm${}^{-1}$	-
ν	motor catalyzed substrate burn rate	∼20 s${}^{-1}$ [17]	$\nu^{\prime }=\frac{\nu }{r}$
$A_{tot}$	total substrate concentration	∼100 μm${}^{-1}$ [16]	$a_{tot}=\frac{A_{tot}}{a_{0}}$
$k_{off}$	substrate unbinding rate	-	$k_{off}^{\prime }=\frac{k_{off}}{r}$
$k_{on}$	substrate binding rate	set to ∼20 s${}^{-1}$	$k_{on}^{\prime }=\frac{k_{on}}{r}$
*K*	substrate equilibrium constant	-	-
ω	bound complex dissociation rate	-	$\omega^{\prime }=\frac{\omega }{r}$
Δ	distance separating bound complex and surface-bound states	-	$\delta =\frac{\Delta }{\sigma }$
ΔG	energy released per substrate molecule	∼15 *K*${}_{B}$*T* [16]	$\Delta g=\frac{\Delta G}{k_{B}T}$

## Data Availability

Simulation code may be accessed at https://github.com/mar-rem/Optimizing-efficiency-and-motililty-of-a-polyvalent-molecular-motor, (accessed on 1 May 2022).

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
