# Peer review of "Optimizing Efficiency and Motility of a Polyvalent Molecular Motor"

_micromachines, 2022, doi:10.3390/mi13060914_

Round 1
Reviewer 1 Report
The authors report an approach to optimize the efficiency and mobility of a polyvalent molecular motors based on a model which entails all simplified relevant kinetics of burnt bridge ratchets. The simplified approach and method behind this paper are suitable and practical for further investigation of a rather complete model. Overall, the manuscript and the results are intriguing and suited for publication in this journal. And there is one issues that needs to be addressed.
- The authors build the hub equation of motion (Eq 1) based on the over-damped assumption. Since the hub is in the proximity of a surface, the mobility and corresponding friction coefficient is much affected by the hydrodynamics close to the surface (e.g. A. Alexander-Katz and R. R. Netz 2007 EPL 80 18001). It would be useful if the authors describe briefly the hydrodynamic effect although they used the simplified picture.
Author Response
The authors build the hub equation of motion (Eq 1) based on the over-damped assumption. Since the hub is in the proximity of a surface, the mobility and corresponding friction coefficient is much affected by the hydrodynamics close to the surface (e.g. A. Alexander-Katz and R. R. Netz 2007 EPL 80 18001). It would be useful if the authors describe briefly the hydrodynamic effect although they used the simplified picture.
We thank the reviewer for highlighting this important point. Indeed, the drag coefficient changes as the hub nears the surface, increasing as it gets closer. Our picture is that the hub is constantly tethered to the surface throughout the motion and is thus at roughly a constant height above the surface. So, although the drag coefficient will be different from what it is in the bulk solution, we take it to be a constant.
We have added the following text at Lines 91-96 and included the reference that the reviewer suggested. "For a spherical particle moving through bulk, this may be approximated as $\gamma = 6 \pi \eta R$, where $\eta$ is fluid viscosity and R is the radius of the particle. However, due to hydrodynamic effects near boundaries, this expression breaks down [24]. Nevertheless, we assume the hub is constantly tethered to the surface during its motion and is therefore at a constant height. So the drag coefficient is constant, albeit larger than what it would be in bulk solution.”
Reviewer 2 Report
Mark Rempel and Eldon Emberly establish a mathematical model to study the efficiency of a polyvalent molecular motor using a burnt bridge ratchet mechanism. By systematically investigating the dependence of the key parameters, they find that an optimal burn rate and substrate concentration can lead to optimal Stokes efficiency of the motor hub. This work deepens our understanding of the efficiency of BBRs. I would recommend its publications if they could address the following concerns.
1. There are too many model variables and parameters, and they are scattered in many places. It would be better if the authors can provide a table to show the definition, unit, typical value, and potential reference of all the model parameters. For example, what represents beta in line 96? Why deltag=15?
2. Line 118 to 119, why can we assume kon~r,? And K should be equal Koff'.
3. Line 149 to 150, the derivation of eqn. (7) is not clear, please clarify it.
4. Figure 3, and Figure 4 A and B, the position of the legend of atot and v' should be switched to match with the figure x-axis.
Author Response
- There are too many model variables and parameters, and they are scattered in many places. It would be better if the authors can provide a table to show the definition, unit, typical value, and potential reference of all the model parameters. For example, what represents beta in line 96? Why deltag=15?
We have created two new tables to summarize the parameters in the model and explain the use of $\Delta g = 15$. These are inserted at Lines 126 and 162.
- Line 118 to 119, why can we assume kon~r,? And K should be equal Koff'.
In principle $k_{on}$ and r are different. Substrate binds to the surface at a rate $k_{on}$ which in principle depends on how the surface was prepared with binding sites. The binding of substrate to the hub, which is given by r, depends on the density of hub motors on the sphere and the ability of these motors to bind the substrate. We assume that the kinetics of the hub with the substrate is of the same order of magnitude as the substrate with the surface. With that assumption, we set $k_{on} = r$ for simplicity. We have clarified this on lines 123-126. We have also corrected the definition of K. - Line 149 to 150, the derivation of eqn. (7) is not clear, please clarify it.
The derivation of equation [7] (now equation [8]) was clarified by including steps for the non-dimensionalisation of the numerator and denominator of the stokes efficiency equation (now equation [7]). Additional steps inserted in lines 156-160. - Figure 3, and Figure 4 A and B, the position of the legend of atot and v' should be switched to match with the figure x-axis.
We apologize for any confusion generated by the figures 3 and 4. However, the axes and legends are correct. The legend values indicate the $a_{tot}$ and $\nu'$ for a given curve. For example, in figure 3 A, the plot shows the velocity as a function of $\nu’$ for different $a_{tot}$.